# Glacial Lake Inventory and Lake Outburst Flood/Debris Flow Hazard Assessment after the Gorkha Earthquake in the Bhote Koshi Basin

**Mei Liu [1,2], Ningsheng Chen [1,*], Yong Zhang [1,2] and Mingfeng Deng [1]**

[1]   Key Laboratory of Mountain Hazards and Surface Process, Institute of Mountain Hazards and Environment, Chinese Academy of Sciences, Chengdu 610041, China; lmeimei90@163.com (M.L.); zhangyongcas@163.com (Y.Z.); dmf@imde.ac.cn (M.D.)

[2]   University of Chinese Academy of Sciences, Beijing 100049, China

*   Correspondence: chennsh@imde.ac.cn; Tel.: +86-13808171963

**Abstract:** Glacial lake outburst floods (GLOF) evolve into debris flows by erosion and sediment entrainment while propagating down a valley, which highly increases peak discharge and volume and causes destructive damage downstream. This study focuses on GLOF hazard assessment in the Bhote Koshi Basin (BKB), where was highly developed glacial lakes and was intensely affected by the Gorkha earthquake. A new 2016 glacial lake inventory was established, and six unreported GLOF events were identified with geomorphic outburst evidence from GaoFen-1 satellite images and Google Earth. A new method was proposed to assess GLOF hazard, in which large numbers of landslides triggered by earthquake were considered to enter into outburst floods enlarge the discharge and volume of debris flow in the downstream. Four GLOF hazard classes were derived according to glacial lake outburst potential and a flow magnitude assessment matrix, in which 11 glacial lakes were identified to have very high hazard and 24 to have high hazard. The GLOF hazard in BKB increased after the earthquake due to landslide deposits, which increased by $216.03 \times 10^6$ m$^3$, and provides abundant deposits for outburst floods to evolve into debris flows. We suggest that in regional GLOF hazard assessment, small glacial lakes should not be overlooked for landslide deposit entrainment along a flood route that would increase the peak discharge, especially in earthquake-affected areas where large numbers of landslides were triggered.

**Keywords:** glacial lake outburst flood (GLOF); debris flow; Bhote Koshi; landslides; Gorkha earthquake; hazard assessment

## 1. Introduction

Many studies have demonstrated that most glaciers are retreating because of global warming and that the meltwater makes an important contribution to the development of glacial lakes in the Himalayas [1–5]. The sudden emptying of these lakes due to dam overflow and moraine or ice dam failure releases large volumes of water and sediment in destructive events called glacial lake outburst floods (GLOF) [6,7]. In the Himalaya region, at least 62 GLOF have been reported, which caused catastrophic destruction and fatalities in downstream regions [8–12]. The peak flood discharge can easily attain tens of thousands of m$^3$/s and travel more than 100 km away [8,13]. Given their high magnitude discharge and long runout distance characteristics, the GLOF impact is sometimes transboundary, especially in the Himalayas. More than 10 GLOF events originated in Tibet, and the catastrophic floods killed hundreds of people and destroyed much infrastructure downstream, causing enormous damage in Nepal and India [14,15]. As a result, GLOF hazard assessment is receiving increased attention from researchers and governments.

In previous GLOF hazard assessment studies, only glacial lakes with an area $>10^5$ m$^2$ or volume $>10^6$ m$^3$ were considered to be risky of outburst [15,16]. However, some small outbursts occurred in the high mountain regions, but are often ignored due to the limited scale of the events or difficult access [17]. Veh [18] detected 10 previously unreported GLOF from Landsat time series in a study area covering only 10% of the Hindu Kush Himalayan region. In addition, glacial lake outburst floods are highly unsteady flows characterized by pronounced changes as they propagate down to the valley [13]. The outburst flood can change from a normal flood to a hyperconcentrated flow or debris flow [19,20], and the volumes and peak discharges can increase several to ten times owing to erosion, slides from lateral slopes, sediment entrainment and bulking process along the flow path [21,22]. As an example, in Norway, a glacial lake outburst flood developed into a debris flow due to erosion and blockage, and the volume increased nearly ten times from 25,000 to 240,000 m$^3$ [23]. Sediment can be entrained by scouring unconsolidated deposits of the channel bed, or eroding landslide and collapse of lateral slopes [24,25]. In the seismic belts, large numbers of weak structures and broken rocks are developed along the active fracture zone, and the soils become looser after an earthquake [26]. Studies showed that large earthquakes, such as the Chi-chi earthquake and Wenchuan earthquake, trigger many collapses and landslides, resulting in an increase in loose deposits [27,28]. Although rare reported GLOF events in the Himalaya are directly triggered by earthquakes [29], the loose deposits and landslides induced by earthquakes may affect the magnitude and impact of GLOF. Therefore, it is necessary to build a GLOF hazard assessment model, considering small glacial lakes and the scenario of glacial lake outburst debris flows after earthquakes, especially in areas where many collapses and landslides have developed along the channels.

The Bhote Koshi Basin across China and Nepal, is a highspot area of glacial lakes and GLOF events (Wang and Jiao, 2015). Four glacial lakes have experienced six GLOF events since 1935 (Figure 1). Taraco Lake failed on 28 August 1935, and the GLOF damaged more than 10 hm$^2$ of wheat fields (Lv, 1999); The Jialongco GLOF occurred on 23 May and 29 June 2002, which caused 7.5 million yuanin economic losses in Nyalam County (Chen et al., 2007). The Cinrenmaco Lake experienced two GLOF events, first in 1964 and second on 10 July 1981. The GLOF in 1981 had the most destructive effects, in which more than 200 people were killed in Nepal, and the total losses were estimated at approximately three million dollars [14,30]. The latest GLOF event occurred on the night of 5 July 2016, which was caused by Gongbatongshaco (GBTSC) Lake in the Zhangzangbo Valley. GBTSC is a small moraine-dammed lake, with a surface area of $1.7 \times 10^4$ m$^2$ and it was almost empty after the outburst. Although it only released $1.1 \times 10^5$ m$^3$ of water, the peak discharge reached 2400 m$^3$/s at Khukundol, 30 km downstream from the lake, due to severe erosion and sediment entrainment [31]. The GLOF caused severe damage downstream of Bhote Koshi, damaging 77 houses, 3 bridges and the Araniko Highway, and destroying the intake dam of the Upper Bhote Koshi Hydropower Project in Nepal (Figure 1). The 2016 GLOF damage sits within the area affected by the Gorkha earthquake (magnitude M 7.8 in 2015), where extensive landslides and rockfalls were triggered on the slopes, and some landslide deposits even blocked the river [32,33]. Therefore, the small glacial lake GBTSC GLOF caused a serious disaster, which caused us to reconsider the small-lake induced GLOF hazard after the earthquake.

The aims of this study are: (1) to establish a detailed glacial lake inventory of BKB after the Gorkha earthquake, based on high resolution remote sensing satellite images; (2) to evaluate the GLOF hazard of BKB considering the scenario that outburst floods evolve into debris flows due to erosion and entrainment of loose solids.

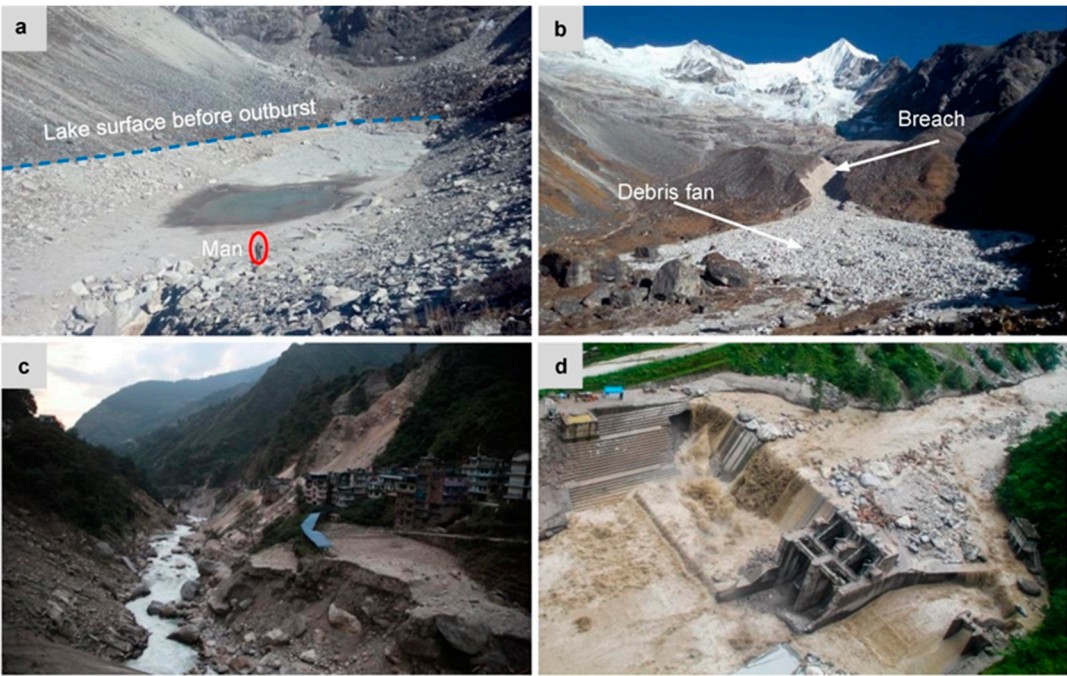

**Figure 1.** Photographs of Gongbatongshaco (GBTSC) Lake and the glacial lake outburst floods (GLOF) damage caused downstream: (**a**) GBTSC Lake after outburst, (**b**) the breach and debris fan in front of the lake, (**c**) landslides and river bank collapse triggered by GLOF near Friendship Bridge and (**d**) the destroyed dam of Upper Bhote Koshi Hydroelectric Project in Nepal.

## 2. Study Area

The study site is located in the central Himalayas and covers latitudes 27°37′–28°31′ N and longitudes 85°40′–86°20′ E with an area of 3406 km$^2$ (Figure 2). Bhote Koshi, which is also called Poiqu in China, is a transboundary river with a length of 143 km. It originates in the Bangbulei Mountains in northern Nyalam County, China, flows into Nepal, and at last feeds into the Ganges River. The Araniko Highway, built along the Bhote Koshi Valley, is a key trade and transport route between China and Nepal. Approximately 200,000 people live in the watershed, among which only 2.7% of them live in China, and the infrastructure in this region is particularly vulnerable [34].

The Bhote Koshi Basin stretches across the Higher Himalaya and Lower Himalaya, and the South Tibetan Detachment System (STDS) and the Main Central Thrust (MCT) pass through it. The basin is strongly affected by seismic activity. According to the statistics of the United States Geological Survey (USGS) earthquake records (http://earthquake.usgs.gov/earthquakes), there were 213 earthquakes (magnitude larger than M 4.5) in the area of 150 km$^2$ around BKB from 1983 to 2016, including a M 8.3 earthquake, three earthquakes larger than M 7.0 and 79 earthquakes equal or larger than M 5.0. The latest large earthquake, M 7.8, on 25 April 2015 and its largest aftershock (M 7.3) on 12 May 2015, produced severe impact in the study area. The epicenter of the major aftershock was only 19 km southeast of Kodari. The lower part from Zhangzangbo Valley to Dolalghat, which is approximately half of the region, was located in seismic intensity zones VIII and VII, and the upper part was in the VI zone according to the seismic intensity of the Gorkha earthquake provided by the USGS National Earthquake Information Center (Figure 2).

The elevation ranges from the highest peak of Mt. Shishapangma at 8012 m to the lowest point of 591 m in Dolalghat, Nepal. Given the large relief, the landforms are different from north to south. In the north and central parts of the basin are alpine regions and gorges, while the valley becomes broader in the south. The climate also varies considerably from south to north. The Himalayan southern slope region of the basin is affected by the Indian monsoon and experiences high precipitation levels. Meanwhile, due to blockage by the Himalayan range, the warm, moist air from the Indian monsoon

can hardly reach the northern part of the basin. According to monthly data obtained from the Nyalam meteorological station (3810 m a.s.l.) and the Zhangmu meteorological station (2250 m a.s.l.), the mean annual temperature ranges from 3.8 °C to 10.1 °C, and the mean annual precipitation ranges from 643.4 mm in the north to 2820.6 mm in the south.

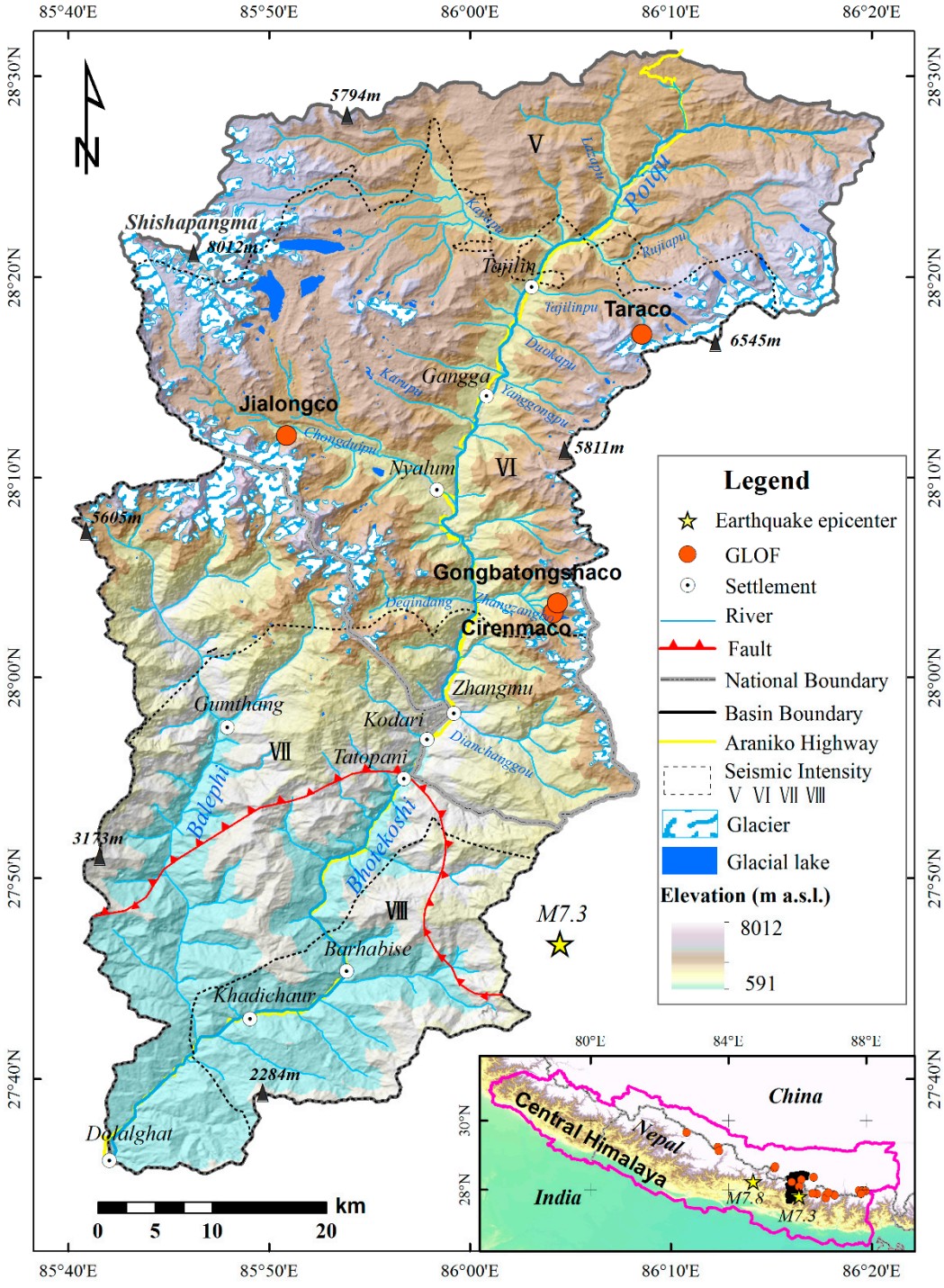

**Figure 2.** Study area and GLOF events in Bhote Koshi Basin.

## 3. Methods

### 3.1. Glacial Lake and Landslide Inventory Mapping

Glacial lake and landslide identification were based on GaoFen-1 (GF-1) satellite images. Twenty-two GF-1 images obtained from the China Center for Resources Satellite Data and Application (http://www.cresda.com/CN/) were Level-1A products, with cloud coverage less than 20% (Table 1). Seven images were used to map glacial lakes and fifteen images were used to map landslides pre- and post-earthquake. Geometric correction and image sharpening were conducted in ENVI 5.2 before mapping in ArcGIS 10.2. The resolution of pan sharpened images was 2 m. This high-quality imagery available allowed us to recognize glacial lakes as small as 0.01 km$^2$. Manual visual interpretation was used to identify glacial lakes and landslides.

**Table 1.** GaoFen-1 images used in this study.

| Data Usage | Sensor | Product ID | Date | Cloud (%) |
|---|---|---|---|---|
| Glacial lake mapping | PMS1 | 2056413 | 20 December 2016 | 13 |
| | PMS1 | 2056415 | 20 December 2016 | 4 |
| | PMS1 | 2056414 | 20 December 2016 | 3 |
| | PMS1 | 1929663 | 1 November 2016 | 13 |
| | PMS1 | 1929664 | 1 November 2016 | 16 |
| | PMS1 | 1929665 | 1 November 2016 | 6 |
| | PMS2 | 1524197 | 14 April 2016 | 15 |
| Landslide mapping post-earthquake | PMS2 | 1242505 | 13 December 2015 | 11 |
| | PMS2 | 1242506 | 13 December 2015 | 1 |
| | PMS2 | 820531 | 22 May 2015 | 1 |
| | PMS1 | 827786 | 23 May 2015 | 3 |
| | PMS2 | 1062062 | 26 May 2015 | 14 |
| | PMS2 | 1062061 | 26 September 2015 | 0 |
| | PMS2 | 1242505 | 13 December 2015 | 11 |
| | PMS1 | 1251892 | 17 December 2015 | 3 |
| Landslide mapping pre-earthquake | PMS2 | 751296 | 11 April 2015 | 0 |
| | PMS2 | 598009 | 19 January 2015 | 12 |
| | PMS2 | 507470 | 9 December 2014 | 1 |
| | PMS3 | 507469 | 9 December 2014 | 19 |
| | PMS1 | 646048 | 22 September 2014 | 16 |
| | PMS1 | 232717 | 22 May 2014 | 2 |
| | PMS1 | 142225 | 30 December 2013 | 9 |

All glacial lakes were verified and modified against Google Earth to see if there are some misinterpretations of the results due to the effect of terrain shadow. The characteristics and surrounding information of all lakes (larger than 0.01 km$^2$) were measured or estimated, aided by Topography Mission digital elevation model (SRTM DEM) (30 m) and Google Earth. These data compose a complete inventory and provide a basis for identifying dangerous glacial lakes. The inventory of the database consisted of 17 parameters, and some important attributes are explained as follows:

(a) Name: some glacial lakes were annotated according to the topographical map of 1978.
(b) Longitude and latitude: the central location of a glacial lake was calculated automatically in ArcGIS based on WGS84 coordinates.
(c) Elevation (m a.s.l.): the central elevation of a glacial lake was derived from the DEM.
(d) Dam type: moraine dam, ice dam and bedrock dam, which was specified based on remote sensing images and the topography map (1:100,000; produced in 1978).
(e) Area (km$^2$): the glacial lake surface area was calculated automatically in ArcGIS 10.2, based on UTM projection zone 48 on a WGS84 ellipsoid.
(f) Dam width (m): these values were estimated using Google Earth.

(g)  Volume (m³): each glacial lake's volume was estimated using Equation (1), which was established between lake areas and volumes of lake water based on data from 33 Himalayan glacial lakes measured in the field [34],

$$V_{gl} = 0.0578 A_{gl}^{1.4683} \tag{1}$$

where, $A_{gl}$ is glacial lake area.

(h)  Estimated freeboard values (1, or 0): the height of the freeboard is difficult to measure by remote sensing but is a crucial parameter that influences dam failure. Here, we estimated whether the height was larger than only a few meters (the value was 1) or indeed close to zero (the value was 0) [35], so it is a semiquantitative parameter.

(i)  Potential triggering impacts: whether the mass movement around a glacial lake can enter into the lake, such as rockfalls (R), landslides (L), ice and glacier avalanches (IGA), debris flows (DF) or flood from a lake situated upstream (ULF). If there is no mass movement, the value was null. This was identified based on Google Earth and the slope maps derived from the DEM, so it is also a semiquantitative parameter.

(j)  Distance to mother glacier (m): the distance between the back edge of a glacial lake to the mother glacier. If they are in contact, the value was 0; if there is no glacier around the lake, the value was set to null.

(k)  Distance to the nearest settlement (m): the drainage distance from the glacial lake dam to the nearest major settlement was measured using ArcGIS 10.2.

(l)  Drainage gradient (°): the average drainage gradient was estimated by a DEM-derived drainage map.

In this study, the term landslide refers to mass movement of a slope, including rockfalls, slope failure and soil slides. Most landslides can be easily identified by visual inspection for vegetation loss or deposits. If there is no vegetation in some areas, the morphology needs careful attention. Landslides are classified as pre-earthquake and post-earthquake landslides. The landslide volume (*V*) was estimated using a power-law landslide area–volume empirical formula (Equation (2)):

$$V_s = \alpha A_s^{\gamma} \tag{2}$$

where $A_s$ is landslide area, $\alpha$ and $\gamma$ are empirically calibrated scaling parameters derived from mixed soil and bedrock landslides in the Himalayas; $\alpha$ is 0.257, and $\gamma$ is 1.36 [36].

*3.2. Glacial Lake Outburst Hazard Assessment*

Glacial lake outburst hazard assessment includes two steps, glacial lake outburst potential assessment and flow magnitude assessment. First, a qualitative method was used to identify glacial lake outburst potential; then, the outburst flow characteristics were determined, flood or debris flow according to loose matter along the flow path and the channel gradient, and then the magnitude at the nearest settlement was calculated. Finally, the GLOF hazard was derived by the glacial lake outburst probability and flow magnitude based on a matrix diagram, which has been widely used in flood, landslide and rock fall hazard assessments [7,37]. GLOF hazards in BKB were divided into four classes: "Very High", "High", "Medium" and "Low". The process of GLOF hazard assessment is summarized in Figure 3.

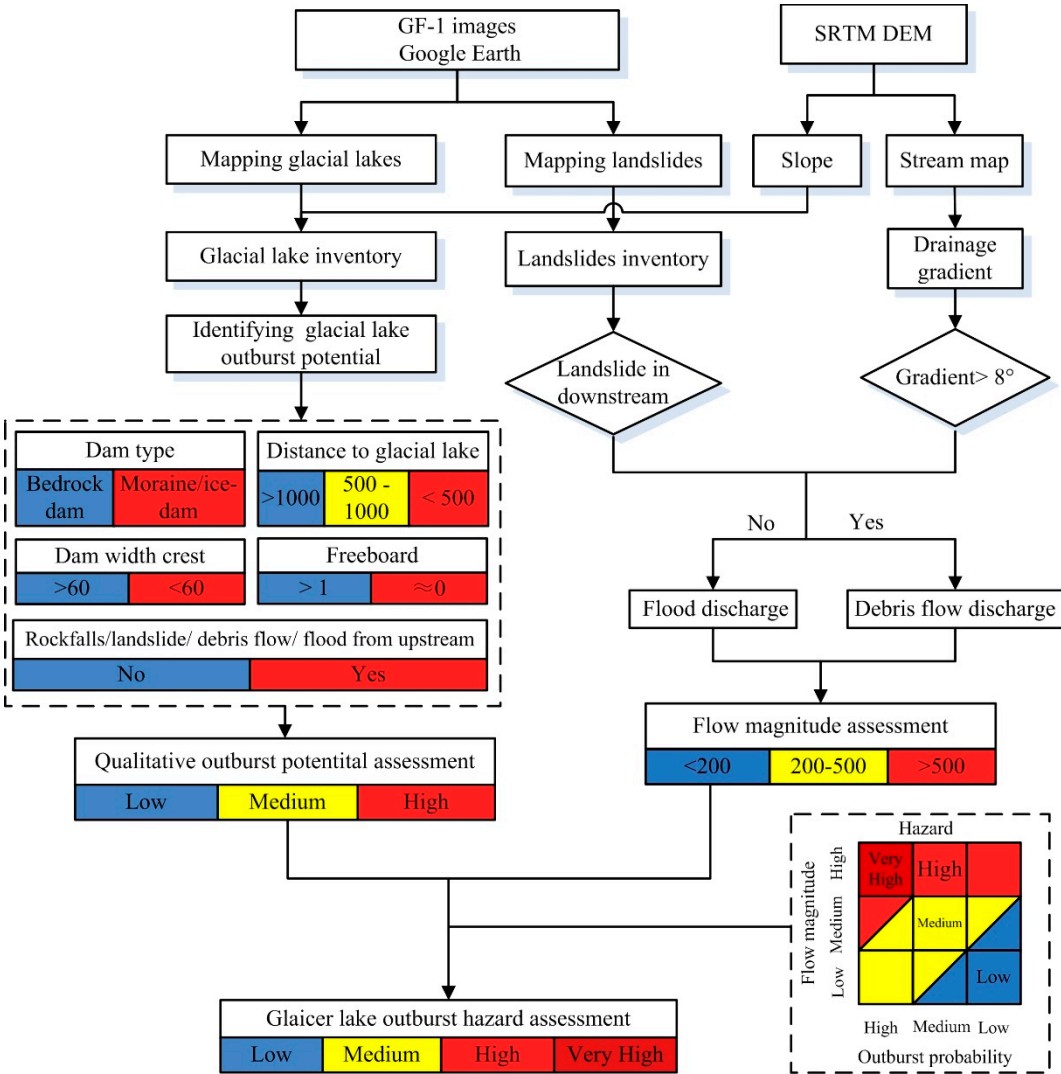

**Figure 3.** Flow chart illustrating the process used for creating the glacial lake inventory of Bhote Koshi Basin in 2016, classifying glacial lake outburst potential and outburst risk assessment.

### 3.2.1. Glacial Lake Outburst Potential Assessment

Many criteria and schemes, derived from GLOF experiences all around the world, have been proposed to assess and identify potentially dangerous glacial lakes based on GLOF [38–42]. Here, a qualitative assessment method is proposed to identify the potentially dangerous glacial lakes from three criteria. The first one is potential triggering impacts, such as rockfalls, landslides, snow and ice avalanches, debris flows and flood from a lake situated upstream [43]. Such mass movement entering the lakes trigger displacement waves that subsequently overtop and erode the dams is the most common cause of dam failure in the Himalaya [8,17,44–46]. Steep glacier surfaces that are in contact with or close to a lake are prone to ice avalanches [47–49]. In addition, steep topography is also likely to cause rockfalls and landslides, and as a glacier retreats, much glacial debris remains, which may start a debris flow under heavy rainfall or intense glacier melting [8,38,50]. The second one is dam stability. Studies show most of the GLOF events in the Himalayas are caused by moraine- or ice dam failures, and the bedrock dams are the more stable with low outburst probability [8,51]. The dam width crest is an indicator for the susceptibility of a dam to fail [52]. The third one is freeboard, which is considered a crucial parameter that influences whether a potential impact wave overtops the dam [7,49].

Five key indicators were selected to identify glacial lake outburst potential according to the three criteria. These parameters of each glacial lake were easily obtained from the 2016 glacial lake inventory database. The critical values for assessment are given in Figure 3. The key indicator was defined with qualitative probabilities high, medium and low, and considered independently. The overall potential is not the mean of the individual indicators. A high-potential outburst glacial lake must satisfy three criteria that are high, a low-potential lake has two or three low criteria and no high criteria and the rest are medium-potential lakes. Finally, three potential degrees were classified as high, medium and low.

### 3.2.2. Flow Magnitude Assessment

The flow magnitude is highly dependent on the peak discharge at the breach and the channel condition [53]. The peak discharge depends on the lake volume and the breach geometry [38]. For a rapid hazard assessment, complex breach processes and flow behavior are beyond the scope. In this paper, the worst breach scenario was assumed, i.e., a full breach that empties the glacial lake water completely. The maximum discharge ($Q_p$) was estimated using the empirical formula (Equation (3)):

$$Q_p = 2V_{gl}/t \tag{3}$$

where $V_{gl}$ is the glacial lake volume, and $t$ is the drainage duration in seconds, which is assumed to be 1000 s [54].

The outburst flood peak discharge increases due to erosion and entrained sediments. Thus, we first needed to judge whether an outburst flood would develop into a debris flow. The average channel gradient and unconsolidated deposits along the channel are key factors that affect whether an outburst flood evolves into a debris flow [13,55]. Erosion is found to occur where the channel gradients exceed 8° [38] and abundant unconsolidated deposits are distributed in the channel and on the slopes [56]. Channel gradients were calculated based on drainage maps derived from the DEM. The unconsolidated deposits include moraine deposits, fluvial and glaciofluvial sediments and landslide deposits. The maximum eroded sediment volume per unit channel length varies from ten to hundreds of cubic meters due to local and regional differences in geology, topography and hydrology of torrent catchments [38,54]. Therefore, it was hard to set a value certain of sediment depth or volume eroded by flood in different channels or basins. A rough assessment was used to estimate the flow magnitude to the nearest settlement. Flood peak discharge was estimated using an empirical equation (Equation (4)) [57,58]:

$$Q_{pl} = \frac{W}{\frac{W}{Q_p} + \frac{L}{VK}} \tag{4}$$

where $Q_p$ is the flood peak discharge m$^3$/s; $W$ is the capacity of the lake, m$^3$; $Q_p$ is the peak discharge at the breach, m$^3$/s; $L$ is the distance from the glacial lake dam, m; and $VK$ is an empirical coefficient equal to 3.13 for rivers on plains, 7.15 for mountain rivers and 4.76 for rivers flowing through terrain with intermediate relief [59], which here we set the value as 7.15.

For an outburst debris flow, the water source is the outburst flood. Therefore, the peak discharge of the debris flow consists of outburst flood discharge and soil particle flow. Blocking was not considered here, so the debris flow peak discharge ($Q_{df}$) can be calculated [60]:

$$Q_{df} = (1 + \varphi)Q_{pl} \tag{5}$$

where $\varphi$ is the increase coefficient of debris flow peak discharge, which can be calculated by:

$$\varphi = (\gamma_s - \gamma_w)/(\gamma_s - \gamma_c) \tag{6}$$

where $\gamma_s$ is the specific gravity of the solid material, g/cm$^3$, and usually determined as 2.65 g/cm$^3$; $\gamma_w$ is the unit weight of water, $\gamma_w = 1$ g/cm$^3$; $\gamma_c$ is the unit weight of the debris flow, g/cm$^3$. Studies show

glacial lake outburst debris flow in Tibet is usually diluted flow [52], and the density is 1.3–1.8 g/cm$^3$. For the convenience of calculation, here we set the average value of $\gamma_c$ as the density of GBTSC outburst debris flow, 1.55 g/cm$^3$.

According to Chinese debris flow prevention and control standards (DZT-0220-2006), a peak flow discharge of more than 200 m$^3$/s is defined as a large hazard. However, the scale of a glacial outburst flood/debris flow is usually larger than that of a rainfall-triggered debris flow [16]. Therefore, in this paper, three flow magnitude classes were established: flow discharge <200 m$^3$/s (low), 200–500 m$^3$/s (medium) and >500 m$^3$/s (high). Finally, the GLOF hazard was derived by the glacial lake outburst probability and flow magnitude based on a matrix diagram, which has been widely used in flood, landslide and rock falls hazard assessments [7,37]. GLOF hazards in BKB are divided into four classes: "Very High", "High", "Medium" and "Low".

## 4. Results

### 4.1. Glacial Lake Inventory

A total of 122 glacial lakes larger than 0.01 km$^2$ with an area of 20.38 km$^2$ were identified based on the GF-1 images from 2016 (see Supplementary Materials). According to the dam type, 84 moraine-dammed lakes with a total area of 16.87 km$^2$ accounts for the largest number and area of all lakes. These moraine-dammed lakes are mainly distributed at 5100–5400 m a.s.l. There are 25 bedrock-dammed lakes that account for 15.3% of the area of all lakes. The average area of a bedrock-dammed lake is 0.12 km$^2$, and are mainly distributed at 4100–4700 m a.s.l. The ice-dammed lakes are the least and smallest, occupying 10.7% and 1.9% of the total number and area. The ice-dammed lakes consist of tiny and small lakes, with a mean area of 0.03 km$^2$, mainly distributed at 5000–5200 m a.s.l. (Figure 4a).

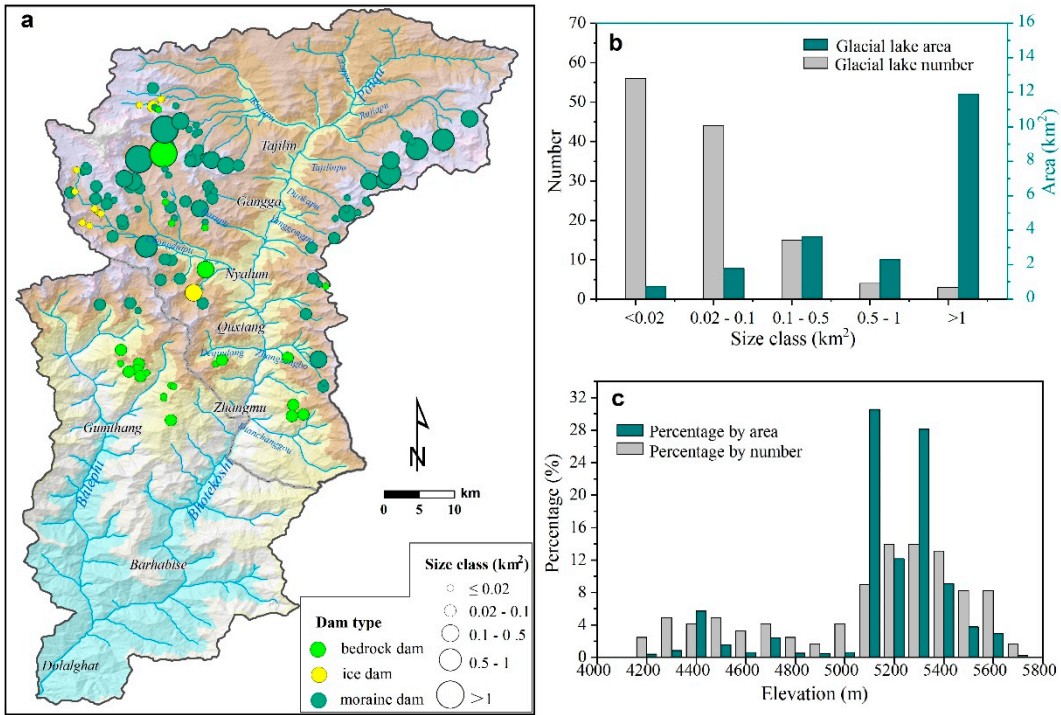

**Figure 4.** Characteristics of glacial lakes in 2016. (**a**) Glacial lakes distribution of different dam type and size class, (**b**) the number and area variation in size classes and (**c**) the percentage of number and area at different elevations.

As the area of glacial lakes vary greatly, from 0.01 to 5.29 km$^2$, we classify them into five size classes: tiny (A ≤ 0.02 km$^2$), small (0.02 < A ≤ 0.1 km$^2$), medium (0.1 < A ≤ 0.5 km$^2$), large (0.5 < A ≤ 1 km$^2$) and giant (A > 1 km$^2$). The percentage of numbers and areas for each size class are shown in Figure 4b.

The main size class of a glacial lake is tiny, accounting for 45.9% (n = 56) of the total number, and small glacial lakes account for approximately 36.1% (n = 44). The total area of tiny and small lakes is 12.4%. Four glacial lakes are large, and three are giant lakes, occupying 69.7% of the total area. Tha mean value of lake area is 0.17 km$^2$, and the largest glacial lake is Galongco, with a surface area of 5.29 km$^2$.

The glacial lakes are distributed at elevations ranging from 4100 to 5750 m a.s.l. and are separated into different elevation classes every 100 m (Figure 4c). Most glacial lakes are located at elevations of 5000–5600 m a.s.l., accounting for 66.4% and 86.6% of the total number and area, respectively. Approximately 27.9% of glacial lakes are located below 5000 m a.s.l. and are evenly distributed in each elevation class with an average 3.5% by number. Approximately 13.9% of glacial lakes are distributed from 5200–5300 m a.s.l. and account for 28.1% of the area of all lakes. It is noticeable that the largest percentage by area is distributed at 5000–5100, which accounts for 30.55%.

*4.2. Glacial Lake Outburst Flood Hazard*

The glacial lake outburst potential assessment results show that 19 glacial lakes have high outburst potential, in which all of these lakes are moraine-dammed and ice/glacier avalanche is the main potential triggering impact; 51 are medium risk, in which two are ice-dammed and nine are bedrock-dammed, and 42 lakes with an area less than 0.1 km$^2$; 52 glacial lakes are low, in which 11 are ice-dammed and 16 are bedrock-dammed (Figure 5a). It is noticeable that 11 out of 19 high outburst potential lakes have an area less than 0.1 km$^2$, and the one bedrock-dammed lake, Gongco, with an area of 2.9 km$^2$, has low outburst potential.

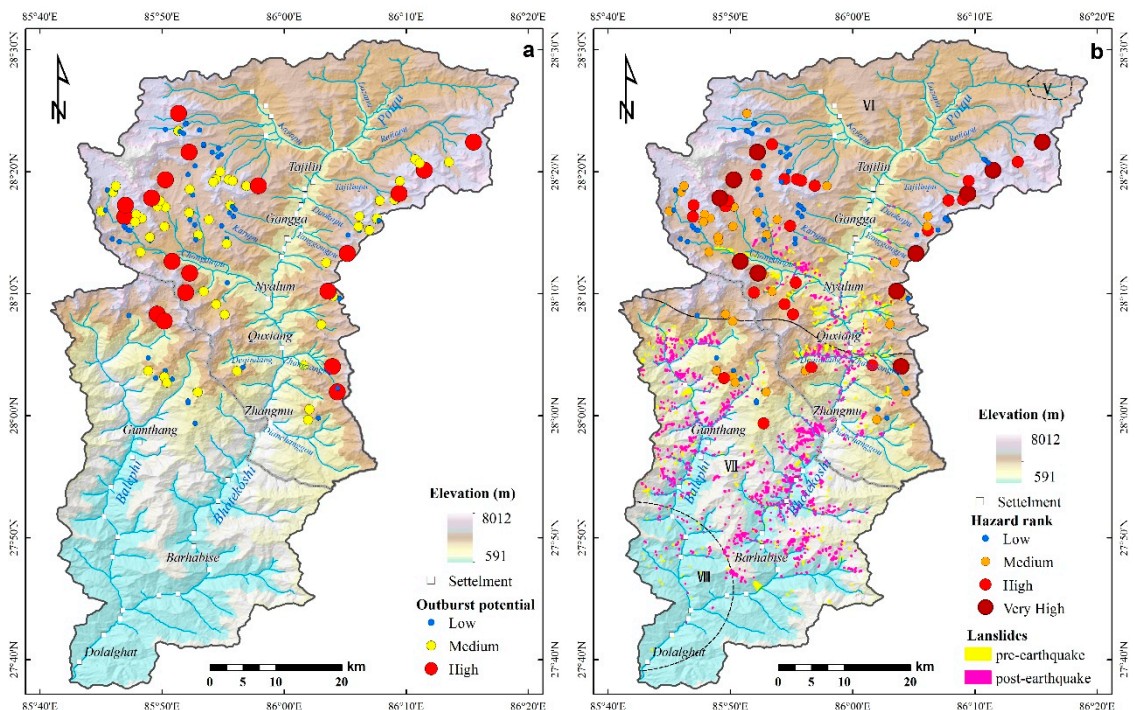

**Figure 5.** (**a**) Glacial lake outburst potential and (**b**) glacial lake outburst flow/debris flow hazard assessed in this study, considering landslide deposit distribution.

In this study, 1670 landslides with a total area of 18.70 km$^2$ were identified, in which 1183, with an area of 12.18 km$^2$, were triggered by the Gorkha earthquake (Figure 5b). These post-earthquake landslides vary in size ranging from 230 m$^2$ to 254,474 m$^2$. Most of the landslides are distributed in the middle and southern parts of the basin, and a large number of Gorkha earthquake-triggered landslides are concentrated in the VII region of seismic intensity. A lot of landslides were distributed in the sub-basins such as Gumthang, Deqingdang, Chongduipu, Zhangzangbo and Dianchangchanggou. Large landslides reach channels, and some small landslides are mostly located on steep slopes

disconnected from a river channel. The other 487 landslides occurred before the Gorkha earthquake and have an area of 6.51 km$^2$. The largest mapped landslide that occurred before the earthquake is 0.81 km$^2$. The total landslide deposit volume is estimated at 91.67 × 10$^6$ m$^3$ before the earthquake, and the volume increased to 216.03 × 10$^6$ m$^3$ after the earthquake. Considering landslide distribution, 73 glacial lake outburst floods are highly prone to debris flow, which will increase the magnitude.

According to the glacial lake outburst potential and flow magnitude, GLOF hazard assessment results are shown in Table 2 and Figure 5. Eleven glacial lakes are identified with very high hazard, among which seven could evolve into debris flows. Twenty-four glacial lakes are high hazard, among which 11 could evolve into debris flows. Thirty-two glacial lakes are identified with medium hazard; the other 55 glacial lakes are considered to have low to no hazard to downstream areas. Four very high-hazard glacial lakes are located in the Chongduipu gully, which presents a large threat to Nyalam County, especially the giant glacial lake Galongcuo that could generate a peak flow discharge of about 224,449 m$^3$/s. Both Jialongcuo and Cirenmacuo have burst out twice before, and are also identified as very high hazard due to high freeboards and hanging glaciers behind the lakes. Eight glacial lakes with low-outburst probability but high-magnitude flow are considered to have high hazard. Among these lakes, Gongcuo and Darecuo are bedrock-dammed lakes and have no potential triggering impacts around the lakes, so they are considered to have a low probability of outburst. However, because of their large volumes, the outburst flows are assumed to be high. 63% (n = 22) of the very high and high glacial lakes' areas are larger than 0.1 km$^2$, and their peak flow discharges were larger than 1000 m$^3$/s. Ten small glacial lakes (area <0.1 km$^2$) identified as high hazard. Three small glacial lakes, No. 16 (area 0.09 km$^2$), No. 18 (area 0.05 km$^2$) and No. 81 (Nongjue, area 0.07 km$^2$), are considered very high hazard for they may cause peak debris flow discharges of 1118 m$^3$/s, 597 m$^3$/s and 994 m$^3$/s at the nearest settlement, respectively.

**Table 2.** Very high and high hazard glacial lakes.

| Id | Name | Longitude (°) | Latitude (°) | Elevation (m) | Dam Type | Area (km$^2$) | $Q_{pl}$ (m$^3$) | $Q_{df}$ (m$^3$) | Probablity of Outburst | Flow Magnitude | Hazard |
|---|---|---|---|---|---|---|---|---|---|---|---|
| 1 | Qiezelaco | 86.26 | 28.37 | 5532 | moraine | 0.26 | 1967 | | High | High | Very High |
| 3 | Cawuqudenco | 86.19 | 28.34 | 5423 | moraine | 0.55 | 6666 | | High | High | Very High |
| 7 | Paquco | 86.16 | 28.30 | 5307 | moraine | 0.58 | 7950 | | High | High | Very High |
| 16 | | 86.09 | 28.22 | 5178 | moraine | 0.09 | 726 | 1814 | High | High | Very High |
| 18 | | 86.06 | 28.17 | 5194 | moraine | 0.05 | 239 | 597 | High | High | Very High |
| 22 | Cirenmaco | 86.07 | 28.07 | 4633 | moraine | 0.34 | 5087 | 12,717 | High | High | Very High |
| 34 | Gangxico | 85.87 | 28.36 | 5212 | moraine | 4.52 | 172,879 | | High | High | Very High |
| 61 | Galongco | 85.84 | 28.32 | 5077 | moraine | 5.29 | 145,746 | 364,365 | High | High | Very High |
| 62 | | 85.82 | 28.30 | 5093 | moraine | 0.27 | 1832 | 4580 | High | High | Very High |
| 80 | Jialongco | 85.85 | 28.21 | 4380 | moraine | 0.63 | 8336 | 20,840 | High | High | Very High |
| 81 | Nongjue | 85.87 | 28.19 | 4628 | moraine | 0.07 | 398 | 994 | High | High | Very High |
| 2 | Youmojiaco | 86.23 | 28.35 | 5337 | moraine | 0.55 | 4881 | | Medium | High | High |
| 6 | Gangpuco | 86.16 | 28.32 | 5543 | moraine | 0.22 | 2355 | | Medium | High | High |
| 8 | Southhu | 86.15 | 28.30 | 5343 | moraine | 0.17 | 1227 | | Medium | High | High |
| 9 | Taracuo | 86.13 | 28.29 | 5257 | moraine | 0.23 | 2186 | | Medium | High | High |
| 10 | Tuzhuocuo | 86.10 | 28.25 | 5201 | moraine | 0.15 | 1309 | 3272 | Low | High | High |
| 23 | | 86.03 | 28.07 | 4486 | bedrock | 0.03 | 257 | 642 | Medium | High | High |
| 32 | Yinreco | 85.89 | 28.37 | 5245 | moraine | 0.28 | 2878 | | Low | High | High |
| 40 | Mabiya | 85.91 | 28.32 | 5384 | moraine | 0.14 | 931 | | Medium | High | High |
| 42 | | 85.92 | 28.32 | 5345 | moraine | 0.08 | 504 | | Medium | High | High |
| 43 | Mulaco | 85.93 | 28.32 | 5306 | moraine | 0.11 | 760 | | Medium | High | High |
| 44 | Xiahu | 85.95 | 28.31 | 5232 | moraine | 0.31 | 3352 | | Medium | High | High |
| 51 | Cuonongjue | 85.92 | 28.26 | 5095 | moraine | 0.23 | 2353 | | Low | High | High |
| 63 | | 85.83 | 28.29 | 5013 | moraine | 0.26 | 1863 | 4658 | Medium | High | High |
| 64 | | 85.83 | 28.29 | 5050 | moraine | 0.06 | 204 | 511 | Medium | High | High |
| 70 | | 85.78 | 28.29 | 5418 | moraine | 0.05 | 130 | 324 | High | Medium | High |
| 72 | | 85.78 | 28.27 | 5309 | moraine | 0.07 | 184 | 459 | High | Medium | High |
| 83 | | 85.87 | 28.17 | 4712 | moraine | 0.04 | 125 | 312 | High | Medium | High |
| 84 | Daroco | 85.92 | 28.18 | 4366 | bedrock | 0.48 | 10,966 | 27,414 | Low | High | High |
| 85 | | 85.91 | 28.15 | 4486 | ice | 0.20 | 2468 | | Medium | High | High |
| 86 | | 85.92 | 28.14 | 4871 | moraine | 0.09 | 597 | | Medium | High | High |
| 88 | | 85.94 | 28.07 | 4524 | bedrock | 0.06 | 391 | 977 | Low | High | High |
| 89 | Bhairab Kunda | 85.88 | 27.99 | 4102 | bedrock | 0.06 | 304 | 760 | Low | High | High |
| 102 | | 85.83 | 28.05 | 4250 | bedrock | 0.07 | 210 | 524 | Low | High | High |
| 103 | Gongco | 85.87 | 28.33 | 5113 | bedrock | 2.09 | 28936 | | Low | High | High |

Note: $Q_{pl}$ is the flood peak discharge at the nearest settlement and $Q_{df}$ is the debris flow peak discharge at the nearest settlement.

## 5. Discussion

### *5.1. Glacial Lake Inventory*

The new 2016 glacial lake inventory indicates that BKB is highly developed glacial lakes. Glacial lake inventory studies have also been conducted in other regions along the Himalayas [61–64]. Glacial lake studies in the Himalayas show that the greatest numbers and areas of glacial lakes are distributed in the central Himalaya [2,7,9,10,62,65]. To compare the glacial lakes and GLOF of BKB with other regions throughout the central Himalayas, the Gyirong River Basin (GRB), which is next to BKB with a similar area was selected and the glacial lake density (glacial lake number/basin area) and lake area per basin area (total glacial lake area/basin area) were calculated (Table 3). The results show that the glacial lake density of BKB is four times that of the central Himalayas and the lake area per basin area is four times that of GRB. The basin area of GRB is larger than BKB, and the glacial lakes density is similar, while the lake per basin area varies greatly. This is due to more large and giant lakes in BKB. According to the statistics, the largest lake is less than 0.5 km$^2$ in GRB, while there are seven lakes larger than 0.5 km$^2$ with the largest being 5.29 km$^2$ in BKB. Studies show the glacial lake expansion rate reaching 0.26 km$^2$/year in Poiqu [66], while the rate of GRB is 0.09 km$^2$/year [63]. Glacial lake expansion is the result of glacier retreating response to climate change. That means BKB is more sensitive to climate change than GRB.

**Table 3.** Comparison of glacial lake and glacial lake outburst flood among Bhote Koshi Basin, Gyirong River Basin and Central Himalaya.

| Region | Basin Area (km$^2$) | Number of Glacial Lakes | Glacial Lake Area (km$^2$) | Glacial Lake Density | Lake per Basin Area |
|---|---|---|---|---|---|
| Bhote Koshi Basin | 3406 | 122 | 20.38 | 0.04 | 0.0060 |
| Gyirong River Basin [63] | 4640 | 148 | 7.12 | 0.03 | 0.0015 |
| Central Himalaya [2] | 280,000 | 1943 | 203.7 | 0.01 | 0.0007 |

The analysis of multitemporal and high-resolution remote sensing images during the compilation of the glacial lake inventories provided a good opportunity to identify previously unreported GLOF events [67,68]. Six unreported glacial lake outburst events were found when we mapped glacial lakes from GF-1 and Google Earth. These glacial lakes have retained typical outburst geomorphic and sedimentological features, such as V-shaped breaches, debris fans and subsequent devastated channels (Figure 6). All of them were moraine-dammed lakes, and their surface areas are 0.01–0.11 km$^2$. Two glacial lakes (Figure 6a,b) are located in Keyapu Valley and the other four (Figure 6c–f) are in Chongduipu Valley. All glacial lakes except No. 31 are fed by glaciers, and the distances to the glaciers are less than 500 m. The V-shaped breach and debris fan of glacial lake No. 86 is the largest, and its mother glacier is thick and hangs behind the lake. The surface area of the glacial lake is 0.09 km$^2$ and the freeboard is much more than one. The rest of the outburst events were small scale and seemed to cause no downstream damage since no erosion was observed in downstream channels. The outburst flood formed a deposition fan at the intersection with the main channel, such as glacial lake No. 31, where the outburst deposition blocked the channel and formed a small lake. Some vegetation has covered the debris fan (glacial lake No. 86) and deposition fan (glacial lake No. 81). It shows that the glacial lake outburst occurred a long time ago. However, the outburst time (year) cannot be determined because of the lack of long-term and high-quality (low cloud cover or high-resolution) data. We traced theses lakes on Google Earth images, and it shows that outburst signs have existed since 1984. As we documented in the literature, we found there was a GLOF event in 1955 in Nyalam, but the record did not mention which lake burst out [30]. Four glacial lakes (Figure 6c–f) are located in the Chongduipu Basin upstream of Nyalam County. As we cannot be sure of the exact outburst time, the outburst magnitudes can be estimated only through debris fans. Glacial lake No. 86 had the largest outburst magnitude with a debris fan area of approximately 252,808 m$^2$, and the gully downstream is highly eroded. This lake may have caused damage in downstream areas. The other three glacial lake deposits

are found at the intersections with the main channel (Chongduipu), which means their outburst floods did not propagate downstream. Therefore, we conclude that the GLOF event in 1955 was caused by glacial lake No. 86.

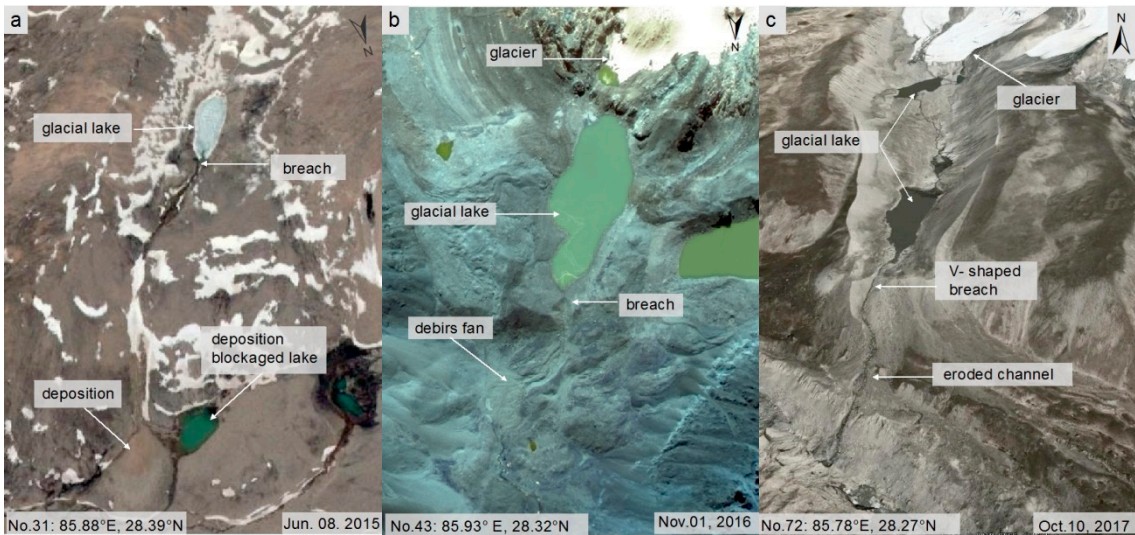

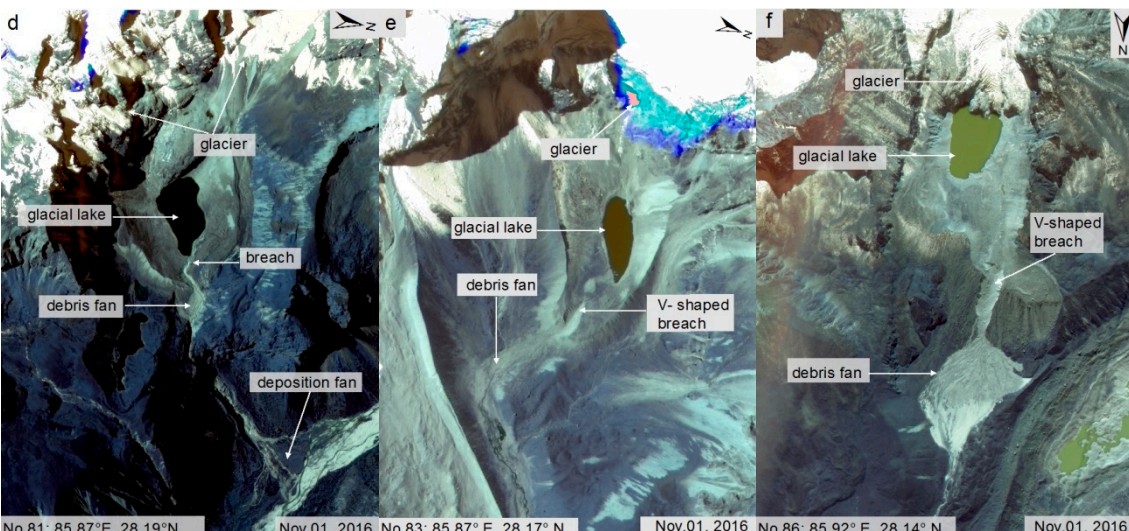

**Figure 6.** Unreported glacial lake outburst flood events were identified based on Google Earth images (**a**,**c**) and GaoFen-1 (GF-1) images (**b**,**d**–**f**) in the Bhote Koshi Basin.

The six undocumented outburst events found show that high-resolution remote sensing images make it possible to trace minor GLOF events that were unreported because of difficult access or few people living in high mountain regions. It also proves that the GLOF frequency is high in BKB. Other unknown and unpublished GLOF have also been found in Bhutan, Nepal and other parts of the Himalayas, based on long-time-series remote sensing data that show glacial lake changes (disappearance or abrupt shrinkage) and typical topographic features, such as exposed debris fans and sediment tails in downstream river channels [9,17,18]. A database of past GLOF events as complete as possible is essential for robust and reliable GLOF hazard assessment [69]. The gradually improving GLOF inventory helps us better understand the mechanism of GLOF and to do hazard assessment.

*5.2. Glacial Lake Outburst Hazard*

A GLOF is a complex process, and the hazard magnitude is determined by the outburst water volume and flood routing [8,22,38]. The outburst water volume is mainly related to the glacial lake volume [54,70]. Since the depth of a glacial lake is hard to acquire by remote sensing data and investigation, the surface area of the lake becomes an important indicator for assessing lake volume and hazard. In previous GLOF hazard assessment studies, glacial lakes smaller than 0.1 km$^2$ were assumed not to pose a hazard potential relevant to downstream locations [47]. Khanal et al. [34] identified 10 critical lakes in the BKB, and all of them are larger than 0.2 km$^2$. Indeed, small glacial lake outbursts can cause damage to downstream locations. According to the inventory of historical glacial lake outburst floods in the Himalayas, small GLOF, such as Zanaco, Geiqu and Choradari Lakes with areas smaller than 0.1 km$^2$, damaged downstream roads and villages [9]. On one hand, a small outburst can create a much larger outburst from another lake located downstream, for examples, the reach of GLOF from lakes Artesoncocha and Chacrucocha [43,67]. On the other hand, a small glacial lake outburst flood can transform into debris flows due to downstream sediment entrainment. For example, a small ice-dammed lake outburst in 2009 (area of 34,000 m$^2$) at Keara in the Andes caused damage 10 km downstream [41].

During the flood routing, landslides on the slopes enter the flood, transforming the flood into debris flows and greatly increasing the discharge, volume and impact. Ignoring the earthquake-induced landslides would underestimate the basin's GLOF hazard. If we do not consider landslides triggered by earthquakes transforming the glacial lake outburst floods into debris flows in the BKB, only nine glacial lakes are identified as having very high hazard, 16 are at high hazard, 12 are at medium hazard and 85 are at low hazard. The hazards rank of two very high hazard small glacial lakes, eight high hazard lakes and 20 medium hazard lakes, accounting for 24.6% of the total lakes, would be decreased. It leads to the GLOF hazard of BKB greatly underestimated.

In this study, the GLOF caused by GBTSC on 25 July 2016 is a good example. GBTSC is located in the Zhangzangbo Valley on the right bank of the Poiqu River, and the average gradient is 182‰ (Figure 7b). This lake was tiny; the surface area before the outburst was 0.01 km$^2$. After the outburst, the lake was almost empty as shown in Figure 7b. The width of the breach was 27 m, and the depth was 9 m. The peak discharge was 618 m$^3$/s, and increased to 4019 m$^3$/s at the section of the Zhangzangbo Valley mouth (approximately 7.2 km from the breach), according to the investigation and assessment report written by the Institute of Mountain Hazards and Environment (http://www.imde.ac.cn). The discharge increased by almost eight times because the outburst flood changed to a debris flow. The Zhangzangbo was in the seismic intensity zone VII and intensely impacted by the Gorkha earthquake. Many landslides were triggered along the river, and some landslide deposits blocked the river (Figure 7c,d). The loose mass volume increased to $8.74 \times 10^6$ m$^3$ in Zhangzangbo according to the landslide distribution. These landslide deposits provided rich masses for the debris flow. Once the GLOF occurred, these deposits were easily eroded and entrained, leading the flood to change to a debris flow and amplifying the discharge. Tens of thousands of landslides were triggered by the M 7.8 (Gorkha) and M 7.3 (Dolakha) earthquakes [71,72]. It will take some time to transport these landslide deposits, which accumulated on the slope or in the channel. Thus, in the region affected by strong earthquakes, we must strengthen the monitoring of high-hazard glacial lakes and pay special attention to glacial lake outburst debris flows after an earthquake.

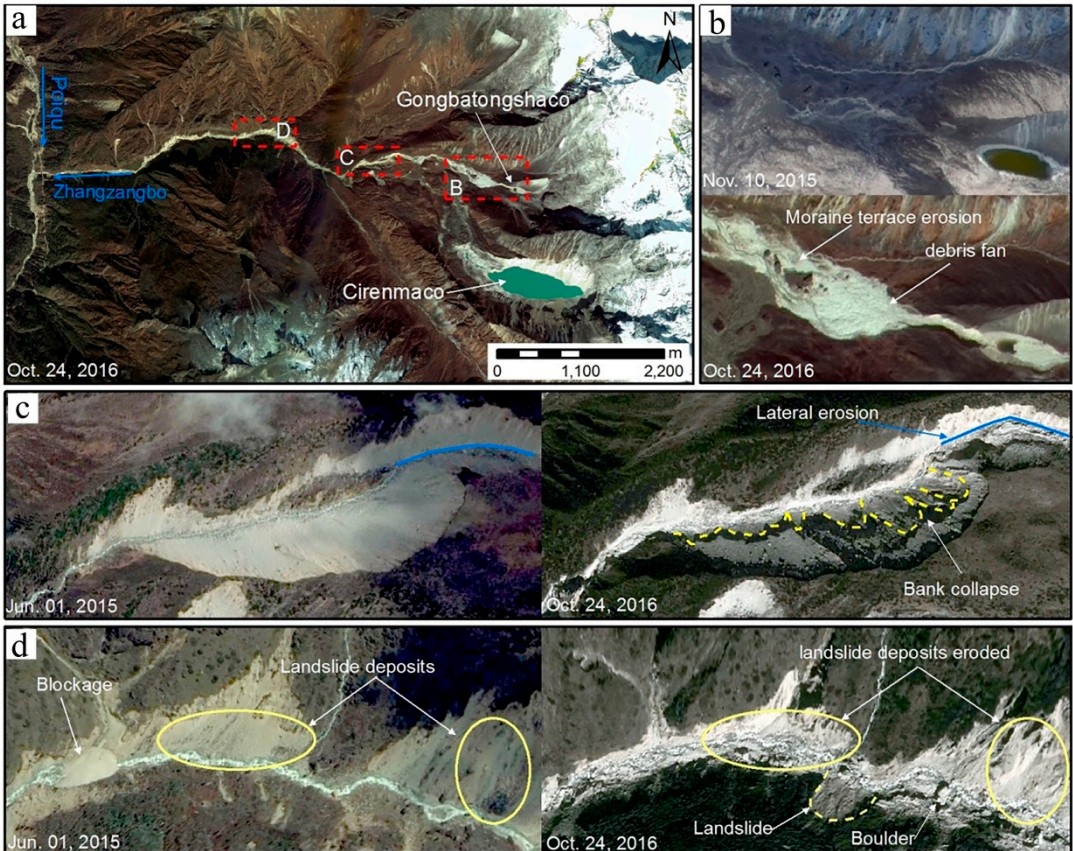

**Figure 7.** Comparison of channel changes before and after GBTSC GLOF along the Zhangzangbo Valley. (**a**) The flow path of GBTSC GLOF; (**b**) GF-1 images showing that the flood left a large debris fan in the front of the lake, and eroded the moraine terrace; (**c**) lateral erosion and bank collapse in the moraine terrace and the width of channel increased; (**d**) landslide deposits distributed along the channel and blocked the channel before bursting, while GLOF eroded landslide deposits and triggered bank slump and landslide afterward, causing an increase in channel width. (Note: the 1 June 2015 images of c and d are from Google Earth, others are GF-1 images).

## 6. Conclusions

In this study a new detailed 2016 inventory of glacial lakes in the BKB was established and six unreported GLOF that occurred before 1984 have been detected with geomorphic outburst evidence based on high-resolution remote sensing images. The BKB is one of the most hotspot small river basins for glacial lakes and GLOF in the central Himalayas. High-resolution remote sensing images are useful for detecting unreported GLOF events in high mountainous regions and sparsely populated regions, which is conducive to improving the GLOF inventory and better assessing GLOF hazard. A rough but more comprehensive method was proposed to assess GLOF hazard, which considers the probability for a flood to develop into a debris flow in the downstream, where large numbers of landslides triggered by earthquake are distributed. The GLOF hazard in BKB increases due to landslide deposits volume, which increased approximately $124.36 \times 10^6$ m$^3$ after the Gorkha earthquake, and 11 glacial lakes are identified as very high hazard, nine are high hazard, 32 are medium hazard and 55 are low hazard. However, about 24.6% of the all lakes' hazards would be underestimated without earthquake-induced landslides, in which most of them are small glacial lakes. Therefore, for regional GLOF hazard assessment, small glacial lakes should not be overlooked for landslide deposit entrainment along a flood route and flood eroding channel bed would increase the peak discharge, especially in earthquake affected areas where large numbers of landslides were triggered. We suggest

that more attention should be paid to the very high and high-hazard glacial lakes and to improving the engineering security standard for defending against flood hazards downstream of BKB.

**Supplementary Materials:** The following are available online at http://www.mdpi.com/2073-4441/12/2/464/s1. Table S1: glacial lake inventory of Bhote Koshi.

**Author Contributions:** N.C. and M.L. conceived the original ideas and drafted the original manuscript. Y.Z. and M.D. carried out field investigation and data collection. N.C. and M.L. revised the original manuscript. All authors have read and agreed to the published version of the manuscript.

**Funding:** This research was funded by National Natural Science Foundation of China (Grant NOs. 41671112 and 41861134008) and the 135 Strategic Program of the IMHE, CAS (Grant NO. SDS-135-1705).

**Conflicts of Interest:** The authors declare there no conflict of interest.

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
