# Peer review of "Glacial Lake Inventory and Lake Outburst Flood/Debris Flow Hazard Assessment after the Gorkha Earthquake in the Bhote Koshi Basin"

_water, doi:10.3390/w12020464_

Round 1

Reviewer 1 Report

The study described in the manuscript "Glacial Lake inventory and Lake Outburst Flood/debris flow Hazard Assessment after the Gorkha Earthquake in the Bhote Koshi Basin" focused on glacial lakes outburst floods hazard assessment in the Bhote Koshi Basin. This catchment is located in the central Himalayas on the territory of Chine and Nepal. Geomorphology of this region and glaciers located in the mountains favour the creation of dam lakes, which due to the geological settings, are intensely affected by the earthquakes. All together increases possibility of catastrophic outburst flood and debris flow which can threaten people.

The manuscript is very interesting. The research is designed appropriate and the methods are adequately described. In my opinion, the results of this study are important for risk analysis and should be used by the local authorities to predict possible risk for local residents.

The authors are using Google Earth products. Unfortunately, I couldn't find in the text if authors have permission to use the Google Earth images. Please add information about licence and permission.

Please write citations and references with the journal rules:
"References: References must be numbered in order of appearance in the text (including table captions and figure legends) and listed individually at the end of the manuscript. We recommend preparing the references with a bibliography software package, such as EndNote, ReferenceManager or Zotero to avoid typing mistakes and duplicated references. We encourage citations to data, computer code and other citable research material. If available online, you may use reference style 9. below. "

https://www.mdpi.com/journal/water/instructions

More detailed comments:

Line 6 and 8: delete "Affiliation 1 Key" and "Affiliation 2 Key".

Line 17: Write the full name of GaoFen-1.

Line 38: "... travel more than 100 km" - is it about distance or speed? Please detail this.

Line 68: should be: 1st and 2nd.

Line 83: Please add "satellite" to the sentence: ... on high-resolution remote sensing SATELLITE images;".

Line 90: Please change the sentence to "(d) the destroyed dam of Upper Bhote Koshi Hydroelectric Project in Nepal".

Line 105: it is written "M7.0" and "M5.0" but in the text is also "Mw 7.8" etc. - please unify.

Line 120 and 121: it is "3810 m.a.s.l." and "2250 m.a.s.l." and should be "m a.s.l.".

Line 121: delete space between numbers and oC.

Line 125: Figure 2. delete "in detail"

Figure 2: Legend: is the yellow star a symbol of the earthquake epicentre? If yes, then add this information to the symbol description.

If the elevation is calculated from the sea level then add "a.s.l.".

Line 136: Change "GF-1" to "GaoFen-1".

Table 1: Which images were used to glacial lake mapping, post-earthquake and pre-earthquake? Mark this in the table.

Line 147: add to the Elevation "a.s.l.".

Line 148: Was the dam type specified in the field or based on remote sensing? Please describe it more specifically.

Line 152: should be "m3".

Line 152: All equations should have numbers. What does mean "A" in this equation and equation from line 169?

Line 155: is the freeboard value dimensionless?

Figure 3: The descriptions in the black squares at the top of the graphic are unreadable.

Line 220: Equation (1) - what does V mean?

Line 237 to 249: Proszę wpisać jednostki w nawiasach kwadratowych []. W tekście autor pisze jednostki na różne sposoby, na przykład m3 / s lub km2 rok-1. Proszę to ujednolicić.

Line 248: space between 1 and unit.

Line 254: space between "flow" and bracket.

Line 271 to 174: add "a.s.l.".

Line 273 and 275: the authors wrote "Approximately". Are the values calculated or estimated? If these values were estimated please write about and give the possible calculation error.

Figure 4: Description of the (a) plot says that this is "the percentage of number and area varies in size of classes". Is it true? The graph shows values, not the percentages.

Line 311: put space between "No.16" and "No.18" and brackets.

Line 315: put space between "No. 72, 81, 88) and brackets.

Figure 5: 5b - Green and dark colour describing landslides is invisible on the map! Consider to make map A and B bigger.

Table 2: Table is cut. Under the table give a description for QPL, QDF and other abbreviations used in Table 2.

Line 342 and 343: Put space between "0.26" and "0.09" and units.

Table 3: Please write the full name of the abbreviations "GLOF", "BKB", "GRB". Tables and Figures should be prepared like a separate complete work.

Line 349: Put space between "events" and bracket.

Line 354 and others: Once the authors write "Glacial lake" and once "glacial lake". Please unify this.

Figure 6 and 7: Put information about sources of images.

Figure 6A: Zoom the investigated area.

Line 383: Put space between "assessment" and bracket.

Line 414: It is "Figure. 7C D". Should be "Figure 7C and D".

Reviewer 2 Report

Dear authors,

First of all you have to clarify what do you mean by the term “glacial lake”(see my comments in the PDF version of your paper).

At some places you should be more precise (see the comments).

Thank you for your contribution,

reviewer

Round 2

Reviewer 2 Report

Dear authors,

I see that you worked a lot to improve the paper. I do not have any comments and I agree with the publishing. Thank you for your effort.

PS: I have only small remark: you mentioned citation  Emmer, Klimeš et al. (2016) in the Answer to reviewers but is not part of the References of the original paper. Is that OK ?

Regards, reviewer